# Flavors of Non-Random Meiotic Segregation of Autosomes and Sex Chromosomes

**DOI:** 10.3390/genes12091338

**Published:** 2021-08-28

**Authors:** Filip Pajpach, Tianyu Wu, Linda Shearwin-Whyatt, Keith Jones, Frank Grützner

**Affiliations:** 1School of Biological Sciences, The University of Adelaide, Adelaide, SA 5005, Australia; filip.pajpach@adelaide.edu.au (F.P.); linda.shearwin@adelaide.edu.au (L.S.-W.); 2Department of Central Laboratory, Clinical Laboratory, Jing’an District Centre Hospital of Shanghai and Institutes of Biomedical Sciences, Fudan University, Shanghai 200032, China; ty_wu@fudan.edu.cn; 3Genome Damage and Stability Centre, School of Life Sciences, University of Sussex, Brighton BN1 9RH, UK; keith.jones@sussex.ac.uk

**Keywords:** non-random segregation, meiotic drive, sex chromosome

## Abstract

Segregation of chromosomes is a multistep process occurring both at mitosis and meiosis to ensure that daughter cells receive a complete set of genetic information. Critical components in the chromosome segregation include centromeres, kinetochores, components of sister chromatid and homologous chromosomes cohesion, microtubule organizing centres, and spindles. Based on the cytological work in the grasshopper *Brachystola*, it has been accepted for decades that segregation of homologs at meiosis is fundamentally random. This ensures that alleles on chromosomes have equal chance to be transmitted to progeny. At the same time mechanisms of meiotic drive and an increasing number of other examples of non-random segregation of autosomes and sex chromosomes provide insights into the underlying mechanisms of chromosome segregation but also question the textbook dogma of random chromosome segregation. Recent advances provide a better understanding of meiotic drive as a prominent force where cellular and chromosomal changes allow autosomes to bias their segregation. Less understood are mechanisms explaining observations that autosomal heteromorphism may cause biased segregation and regulate alternating segregation of multiple sex chromosome systems or translocation heterozygotes as an extreme case of non-random segregation. We speculate that molecular and cytological mechanisms of non-random segregation might be common in these cases and that there might be a continuous transition between random and non-random segregation which may play a role in the evolution of sexually antagonistic genes and sex chromosome evolution.

## 1. Delving into History: The First Evidence of Random Segregation of Chromosomes at Meiosis

The history of chromosome segregation dates back from the time when the link between Mendelian genetics and behavior of chromosomes during meiosis was formed based on the studies of Walter Stanborough Sutton in 1903. Already knowing that homologous chromosomes form pairs, Sutton observed segregation of such homologs during meiosis in the grasshopper *Brachystola*. Although he misidentified the second meiotic division as the reducing one, he claimed that the orientation of homologous chromosomes in the equatorial plate is “purely a matter of chance” [1]. However, as he also acknowledged, he brought no definitive evidence due to the inability to reliably distinguish between the paternal and maternal chromosomes of any given autosome pair. Ten years later, Estella Eleanor Carothers came up with the cytological evidence of what had been proposed by Sutton before—that is, if genes are to be assorted independently, chromosomes bearing these genes should segregate randomly. After the discovery of a single heteromorphic autosome pair in grasshopper, Carothers carefully observed distinguishable heteromorphic chromosomes during the first meiotic division in *Brachystola*. She noticed that one heteromorphic homolog segregates together with the accessory (sex) chromosome with approximately equal frequency compared to the other homolog (Figure 1) [2]. These results fundamentally led to the establishment of what has become a dogma in biology–chromosomes segregate randomly at meiosis. At the time when the first evidence of non-random segregation of the segregation distorter in *Drosophila* was discovered there were no powerful tools to study the relationship between chromosome segregation and factors that control and influence this segregation. In addition to that, this dogma was in agreement with the Mendelian laws–something more easily demonstrated than physical chromosome movement. Now, over 100 years later, there is increasing evidence and understanding of the factors that control random segregation and these are subsequently discussed.

## 2. Building Blocks of Chromosome Segregation

In order to successfully divide genetic information to daughter cells, chromosomes undergo a complex process of segregation. During cell division, kinetochores must form on centromeres of each chromosome. After kinetochores are assembled, chromosomes now possess a structure thanks to which they can be attached to microtubules of a spindle apparatus, which undergoes the process of its own assembly. Once kinetochores of all chromosomes are attached to spindle microtubules, a multistep journey of chromosomes to daughter cells can commence. Chromosome segregation is a highly regulated process occurring during both mitotic and meiotic divisions and involving many other components, including microtubule organising centre (MTOC), components of sister chromatid cohesion and pairing of homologous chromosomes, and much more. Individual components are subsequently discussed in relation to how they act to ensure that chromosomes are segregated to daughter cells.

### 2.1. Centromeres—Not Just Construction Sites for Building Kinetochores

The centromere is an important structure for proper chromosome separation ensuring that only one copy of each chromosome segregates to each daughter cell during cell division. The centromere was initially observed in 1882 and termed in 1936, after that its crucial functions in cell division were widely investigated [3]. Structurally, the centromere is composed of both nucleic acids and proteins—evolutionary conserved histones, as well as non-histone proteins. The centromeric DNA provides a base for the assembly of the centromeric proteins (CENPs) during the entry into meiosis or mitosis [4,5,6]. The interaction between centromeres and microtubules was initially observed by electron microscopy [7,8,9].

A notable effort has been put into identifying centromere DNA sequences in various organisms, but no universal centromere sequence has been found. On the contrary, centromeric DNA between various organisms differs significantly [10]. In mouse, the centromeric region is defined by euchromatin epigenetic markers and repetitive centromeric DNA sequences, which are enriched in minor satellite repeats (600 kbp of 120 bp sequences) (Figure 2) [4]. Minor satellite DNA is the prominent site for CENP-A and CENP-B assembly [4,11]. However, if the minor satellite region is not big enough, CENP-A can also be formed *de novo* elsewhere and its localization is not strictly sequence dependent [12,13,14,15]. CENP-B is a minor satellite DNA binding protein. Despite the centromere sequence variety across species, the CENP-B box (17 bp DNA) is highly conserved between rodents and primates [3]. The mouse pericentromeric region contains inactive chromatin and is composed of major satellite repeats (6 Mbp of 234 bp monomers) (Figure 2) [4], which is required for heterochromatin formation [16,17]. However, the role of major satellite DNA in cell division is still unclear. Pericentric regions act as insulators for the centromere and are epigenetically distinct due to specific histone modifications. Unlike centromere-specific CENP-A, pericentromere contains histone variants H3.3 and H2A.Z. Pericentric histones are generally hypoacetylated and exhibit some methylation marks on histones H3 and H4 that cannot be found in centromeric region [18].

The DNA sequence of centromeres does not provide centromeric identity—at present, it is generally accepted that epigenetic characteristics provide centromeric identity and navigate kinetochore components to centromeres, with CENP-A protein (centromeric variant of histone H3) being particularly important. Experiments involving knock-out of CENP-A in chicken DT40 cells seem to support this claim—CENP-A depleted cells fail to recruit some inner kinetochore components (as well as spindle assembly checkpoint proteins) to centromeres, leading to delayed mitotic progression, chromosome mis-segregation and ultimately cell death during a following G1 stage of the cell cycle [19]. Although CENP-A devoid kinetochores partially maintain their function, this work demonstrates that assembly of kinetochores is a stepwise process with CENP-A as a critical upstream component acting as a link between centromeres and kinetochores. Likewise, studies in *Drosophila* revealed that CENP-A serves as a prominent epigenetic mark allowing for kinetochore formation and microtubule binding on extrachromosomal plasmid constructs and also directs its own incorporation, which leads to self-maintaining of centromere identity through each cell division [20]. Similarly, it has been shown that in *Arabidopsis thaliana* knockdown of CENP-A disturbs chromosome segregation at meiosis leading to unequal number of chromosomes in daughter cells, high frequency of lagging chromosomes, and increased formation of micronuclei in pollen tetrades. This is one of the few studies showing effects of CENP-A depletion on meiosis rather than mitosis and apparently, N-terminal domain of CENP-A is required for its deposition to meiotic chromosome centromeres [21]. Finally, human studies in CENP-A depleted cell culture models suggest that CENP-A is necessary to suppress centromeric recombination and rearrangements to maintain stability [22]. These studies indicate that centromere is not simply a landing pad on which kinetochore can assemble, but it actively directs the process of kinetochore assembly.

The epigenetic foundation of centromeres is further highlighted by the existence of neo-centromeres, which are ectopic centromeres devoid of repetitive DNA that is usually found in centromeres. In humans neo-centromeres are fully functional, enabling cell division upon disruption of the endogenous centromere through ectopic CENP-A deposition [23]. However, neo-centromeres in humans appear to be generally detrimental and are associated with at least two types of cancer, suggesting they are biologically important beyond the level of an individual, such as for karyotype evolution and speciation [24]. Furthermore, holocentromeres are centromeres that span the entire chromosome length and are also defined epigenetically rather than based on sequence. They are organized as a set of individual point centromeres scattered across chromosomes, with each containing only one nucleosome lacking any centromere-specific repetitive sequence [25].

It has been shown conclusively that even though centromeric DNA itself is not sufficient for kinetochore assembly and does not provide centromeric identity *per se*, centromeric transcripts generated from centromeric DNA might be required for accurate loading of CENP-A and also to stably localise CENP-C protein to centromeres, which is a step essential in kinetochore formation [10]. Indeed, it appears that centromeric transcription is essential for proper centromere function including the kinetochore attachment in at least two ways—through chromatin remodelling triggered by the action of RNA polymerase II and production of centromeric RNA transcripts (cenRNA). In the case of the former, results from human and *Drosophila* indicate that the physical passage of RNA polymerase II and its associated transcription factors and other proteins is required to destabilize histone H3-containing chromatin and establish appropriate epigenetic environment in order to deposit CENP-A histone variant that will initiate kinetochore formation [10]. In the case of cenRNA production, it has been shown in fission yeast *Schizosaccharomyces pombe* that cenRNA directly interacts with CENP-A and CENP-C and is also a subject of RNA interference. In this case the processed RNA transcripts recruit the Ago1 protein which associates with the ClrC complex containing Clr4 methylase. This results in the formation of key methylation histone marks that are required to recruit the same protein complexes again (resulting in recurring nucleation), as well as new proteins (such as Swi6, Epe1, Clr3 and others) to further regulate this process [26]. In mammals, the key methylation marks are produced by methylase SUV39H1 and RNA interference is most likely involved too, but the process is less understood in mammals. Importantly, it is the transcription of centromeric regions containing euchromatin markers that is responsible for maintaining the functionality of centromere. On the other hand, the transcription of pericentromeric regions containing inactive chromatin is suppressed due to the presence of distinct epigenetic markers, which prevents transposon reactivation, carcinogenesis and disease [27]. Taken together, centromeric transcription and cenRNA are indispensable for proper centromere function and dysregulation of these results in chromosome instability or cancer [28].

### 2.2. Kinetochores—Mediators of Microtubule Attachments to Centromeres

Kinetochores are multiprotein structures providing link between spindle microtubules and chromosomes and are involved in the early stage of chromosome segregation. They are formed precisely at centromeres and while the presence of a centromere is normally essential for this formation, centromeric DNA sequence itself is not sufficient to determine the position of kinetochore formation, perhaps with an exception of the budding yeast, which possesses very short well-defined centromeric sequences [29]. Indeed, an instructive example is provided by human neocentromeres which are, contrary to standard human centromeres, devoid of repetitive satellite DNA and generated outside of heterochromatic regions in otherwise healthy people. Nevertheless, such neocentromeres are perfectly capable of binding kinetochore components and therefore mediating microtubule attachment [30]. Furthermore, many plant studies support the idea that centromere sequence *per se* is not the key for the kinetochore formation [31].

In vertebrates, kinetochores are composed of more than 100 proteins, which together form inner and outer layers of kinetochore (mediating contacts with chromatin and microtubules, respectively) controlled by other regulatory proteins [32,33]. One of the critically important group of protein complexes involves the so-called constitutive centromere-associated network, or CCAN—together with specialised centromeric chromatin marked by histone CENP-A, these proteins participate in formation of the inner layer of kinetochore (Figure 3). Many proteins participate in the formation of CCAN, including subcomplexes CENP-C, CENP-H-I-K-M, CENP-L-N, CENP-O-P-Q-R-U, and CENP-T-W-S-X. Based on CCAN, the outer kinetochore is assembled to bind microtubules directly and transduces the force required to move chromosomes during metaphase and anaphase. Its core is composed of 10-subunit complexes, termed KMN (Knl1 complex, Mis12 complex, Ndc80 complex) [34,35,36,37]. Additionally, more proteins are involved in the kinetochore organization at meiosis, such as cohesins [38].

As the primary microtubule binding site, the Ndc80 complex interacts with microtubules through its N-terminal regions (Ndc80 and Nuf2) and mediates kinetochore interactions with its C-terminal regions (Spc24 and Spc25) [39,40,41,42]. The Mis12 complex acts as an interaction hub to promote KMN assembly at the inner kinetochore, binding to CENP-C and CENP-T. In addition, it can bind to both the Ndc80 complex and the Knl1 complex, connecting them together [43,44].

Collectively, these proteins together with the KMN complex provide the necessary foundation for microtubule attachment [45,46]. Once a kinetochore is assembled at the centromere, this interface is secured and stable for the entire duration of anaphase. On the contrary, the kinetochore-microtubule interface is more dynamic, with microtubules being able to constantly depolymerise at their plus end even though they are connected to kinetochore. The forces exerted by microtubule depolymerisation are balanced by the forces in the opposite direction and these are usually developed by biorientation of sister kinetochores, which creates tension. This tension is in turn sensed by proteins that contribute to the attachment between kinetochores and microtubules [33]. Hence, the connection between kinetochores and microtubules appears to be dynamic—yet kinetochores are still able to maintain this connection for the duration of anaphase. There are currently two models explaining this phenomenon, conformational wave and biased diffusion. In the former model, disassembling microtubules bend backwards and by doing so, they push on the putative ring structure that is built around a microtubule, which in turn pulls kinetochore towards the minus end of microtubules [47]. In the latter model, kinetochores form many individual attachments with microtubules that diffuse along protofilaments, securing directional movement towards the minus end of microtubules [48]. At present it is still not clear which model is the correct one, but there are studies indicating that some aspects of both models may be valid [49].

### 2.3. The Spindle Apparatus—A Cytoskeletal Machine Segregating Chromosomes

Chromosome segregation depends on the formation and function of spindle apparatus during both mitosis and meiosis. Chromosomes must attach to spindle microtubules in order to be segregated to daughter cells. These attachments are provided by kinetochore microtubules, which together with astral and polar microtubules, hundreds of proteins, and chromosomes themselves form the structure of spindle. There are currently several models describing the process of spindle formation [50]. In particular, model proposing self-organisation of spindle based on the Ran GTPase gradient within a cell is well supported in animals [51,52,53], yeasts [54], and plants [50]. Specifically, after the nuclear envelope breakdown a gradient is formed, with Ran-GTP in the vicinity of chromosomes and Ran-GDP on the cell periphery, which itself is sufficient for spatial coordination of self-organisation of the microtubule-chromosome system and centrosome is not needed [52]. Indeed, it is only logical to assume that such centrosome-independent pathway is likely to be wide-spread and evolutionary conserved since plants lack centrosomes. In addition to that, spindle formation is much faster during meiosis II and centrosome pathway would not be able to “catch up”. In support of this, it was shown that the Ran gradient pathway seems to be even more critical in meiosis II compared to meiosis I [53].

Furthermore, spindle microtubules must capture kinetochores and they do so laterally (chromosomes are captured along microtubule walls), which increases the area of microtubule—kinetochore interface. Once established microtubules and kinetochores form a more stable and firmer end-on attachment (chromosomes are brought to microtubule ends). In *Saccharomyces cerevisiae* the initial capture requires many protein complexes, including already mentioned NDC80 [54]. Once microtubules are attached to kinetochores, tension that is counteracted by sister chromatid or homologous chromosomes cohesion is developed. Such tension is critical for stabilising these attachments [55], complementing the indirect function of Aurora B kinase [56].

The key step of chromosome segregation occurs during anaphase, when chromosomes are segregated to future daughter cells by collective action of microtubule depolymerisation and molecular motors—in other words, chromosomes move poleward during anaphase. The traditional view on the spindle is that kinetochore microtubules (which in higher eukaryotes form bundles of about 20, referred to as K-fibers) directly connect kinetochores to spindle poles, providing a direct mean for their segregation [57]. However, this is a rather simplified view, since such direct connection is not always necessary. Indeed, in some marsupial and eutherian cells kinetochores of chromosomes are not always directly connected to spindle poles–instead, their K-fibers terminate prematurely, never reaching centrosomes. Interestingly, such K-fibers can attach to neighbouring K-fibers or even non-kinetochore microtubules, safely transporting chromosomes to opposite poles thanks to the action of the molecular motor dynein [58]. The very similar situation has been also reported in *Caenorhabditis elegans* [59]. These examples only strengthen the idea of molecular motors being critically involved in chromosome segregation together with depolymerisation of microtubules.

### 2.4. Role of Microtubule Organising Centre (MTOC) and Centrosome in Chromosome Segregation

Although at present it is clear that assembly of the spindle is characterised by a remarkable ability to self-organise, the importance of the MTOC in chromosome segregation is unquestionable. Various groups of eukaryotes, cell types or even same cell types under different conditions possess their own characteristic MTOC in terms of structure and functionality. Generally, MTOC in yeasts is referred to as the spindle pole body (SPB), whereas in animals it usually consists of centrosomes containing centrioles. Contrary to that, plants do not possess centrosomes nor an SPB—instead, their MTOC is mainly formed by γ-tubulin (a ubiquitous protein found in all eukaryotes) and rely more on self-organising capabilities. However, there is one thing all of these groups have in common—their MTOCs serve as an anchoring point for microtubules during chromosome segregation, which is necessary for the proper spindle function regardless of how microtubules nucleate [60]. Indeed, current research demonstrated that the idea of a bipolar spindle with a single centrosome at each end is an oversimplification and although many organisms or cell types do rely on such organisation, centrosomes likely arose in evolution to be responsible for different functions. Indeed, females of many species, including mouse, naturally lack centrosomes in meiotic spindle. In mouse MTOCs are acentrosomal and exhibit unique self-organising capabilities – during meiosis, more than 80 MTOCs come together to form poles of the spindle, functionally replacing centrosomes [61]. Likewise in the fungus gnat *Sciara*, unfertilized embryos lack centrosomes, as well as (contrary to mice) astral microtubules [62].

If MTOCs are perfectly capable of distributing chromosomes to daughter cells together with microtubules and kinetochores, but without centrosomes, which is the case in plants and several animal lineages, why did centrosomes become the part of MTOCs in the majority of animal cells? Although in unfertilised *Sciara* embryos lacking centrosomes chromosomes are segregated normally, the distance between newly formed daughter nuclei is reduced, leading to the formation of abnormal cell arrays after collision of neighbouring nuclei. Thus centrosomes are required for the proper nuclear spacing in this animal [62]. Furthermore, failure of centrosomes to separate before the nuclear envelope breakdown leads to chromosome mis-segregation, suggesting that centrosomes may scan cellular space to increase fidelity of chromosome segregation [63]. Taken together, MTOC itself is another key component securing successful chromosome segregation, whether it contains centrosomes or not.

### 2.5. Cohesins—Protein Complexes Linking Cohesion, Synapsis and Segregation of Chromosomes

Faithful segregation of chromosomes would not be possible without proper cohesion of sister chromatids and homologous chromosomes both in mitosis and meiosis. Cohesins are protein complexes responsible for this crucial task and they consist of four subunits as follows: two SMC proteins (always SMC3 and mitosis or meiosis specific isoform of SMC1), a kleisin protein (Scc1 or its orthologs like Rec8, Rad21 or Rad21L), and Scc3 protein (or its orthologs like SA or STAG proteins) [64,65]. During mitosis, sister chromatid cohesion mediated by cohesins localised around centromeres acts as the counteracting force to the microtubule forces to make sure that sister chromatids are held together until anaphase. By the onset of anaphase, however, this cohesion must be interrupted so it allows chromatids to separate from each other, which is also true for the second meiotic division. In contrast, cohesion around centromeres must be protected in the first meiotic division (the task performed by shugoshin—the protein also responsible for protection of synaptonemal complex at centromeres during prophase I and promoting disjunction of achiasmatic homologs) and instead, cohesin complexes are removed from chromosome arms [66,67,68]. Proteolytic activity of the separase is required for removal of cohesins from centromeres in mitosis and meiosis II and chromosome arms in meiosis I—indeed, oocytes without a functional separase do not segregate chromosomes or extrude a polar body at all. Additionally, cohesins are also removed from chromosome arms in mitosis in the process called prophase pathway, which requires separase, too [69,70]. The importance of cohesins in chromosome segregation is further highlighted by studies indicating the association of reproductive aging of a mother and the decrease of cohesin both on chromosomes arms and centromeres. This has been proposed to lead to a higher risk of aneuploidies in children of older mothers due to cohesin depletion, even though it is still unclear which cohesin subunits are directly affected [66,71].

In addition to pairing their sister chromatids, homologous chromosomes must also pair with each other in order to recombine and properly segregate at meiosis. They do so by forming synapsis through action of both cohesins and synaptonemal complex proteins, which results in homologs (but not sister chromatids) forming bivalents and being recombined and physically linked by chiasmata. The early role of recombination is important for promoting homologous pairing prior to the formation of synapsis. In most organisms, meiosis is indeed chiasmatic and usually at least a single chiasma per bivalent is essential to correctly position bivalents which is in turn necessary for correct chromosome segregation. This does not occur in case of organisms exhibiting achiasmatic meiosis, such as sex chromosomes of marsupials (where the structure called dense plate secures pairing of completely non-homologous sex chromosomes) [72], *Drosophila* males [73], mygalomorph spiders [74] or scorpions [75]. Synapsis is mediated by synaptonemal complex composed of central element containing synaptonemal complex protein 1 (SYCP1) and lateral elements containing SYCP2 and SYCP3, as well as cohesins. The central element is connected to lateral elements by SYCP1. Asynaptic plant, animal and human mutants have been identified—a spontaneous or induced disruption of synaptonemal complex genes can cause asynapsis or desynapsis leading to lagging or completely failing meiotic progression, decreased fertility or infertility and disease [76,77,78]. Similarly, mutations of meiotic cohesins can severely affect synapsis—mutation of *Stag3* causes decrease in levels of all other meiotic cohesins, early prophase I arrest and apoptosis in male and female germ cells [79]. Mutation of *Rad21L* largely impairs homolog association independently of double-stranded breaks [80]. Mutation of *Rec8* results in pairing of sister chromatids during prophase I rather than homologous chromosomes (leading to infertility of males and females), indicating that REC8 is responsible for limiting synapsis of homologs only [81]. Taken together, synapsis mediated by both synaptonemal complex proteins and cohesins is an essential step required for successful meiotic progression and chromosome segregation. Indeed, unsynapsed chromatin presents the risk of triggering a pachytene arrest. Transcriptional silencing of unpaired DNA during meiosis can be found in different organisms and is achieved by different mechanisms [82,83]. Heteromorphic sex chromosomes, such as male sex chromosomes of the XY sex chromosome system in mammals and ZW in birds, are naturally largely non-homologous and therefore do not pair or recombine except for their shared pseudo-autosomal region. In therian mammals this results in the formation of the sex body and transcriptional inactivation through the process of meiotic sex chromosome inactivation (MSCI). Interestingly, in birds and monotremes MSCI appears to be missing [84,85,86]. Incomplete synapsis and MSCI both pose challenges for chromosomes segregation at meiosis [87].

## 3. Chromosomes That Break the Rule of Random Segregation

So far, the building blocks essential for chromosome segregation have been described in terms of generally accepted random segregation. In the following sections, these blocks (that is centromeres, kinetochores, MTOCs, spindles, and cohesins) and possible mechanisms are discussed in relation to the non-random segregation of both autosomes and sex chromosomes.

### 3.1. Meiotic Drive as a Form of Non-Random Segregation of Autosomes

Mendel and later Sutton and Carothers postulated that heterozygotes should transmit alleles to gametes with equal probability and thus chromosomes should segregate randomly for that to happen. We now have an increasing evidence that this is not the case when various processes such as meiotic drive take action [88]. Meiotic drive is the process when a heterozygous individual fails to transmit different alleles (or chromosomes) with equal frequency because one of these alleles (or chromosomes) enhances its own transmission over another [89]. Thus, meiotic drive is a mechanism that can lead to biased segregation of homologous chromosomes. One of the first examples of meiotic drive are the *t*-haplotype in house mouse and segregation distorter (SD) in *Drosophila*, both being examples of the male meiotic drive. In case of male meiotic drive, sperm that do not bear a driving allele or chromosome are usually eliminated or impaired [90]. However, recent research tends to focus on the meiotic drive of chromosomes that occurs in females (therefore often referred to as female meiotic drive). Female meiosis is naturally asymmetric with the extrusion of a polar body, which is genetically a dead end since it is eliminated during meiosis along with all chromosomes it contains. Driving chromosomes (or alleles) can exploit this asymmetry to preferentially segregate to an egg and not to a polar body. Therefore, both types of meiotic drive involve a *trans*-acting driving allele at one locus that impairs transmission of a sensitive allele at another locus. The relationship between these loci is critical as the driver must not recombine onto the sensitive chromosome as this would result in the driver acting against itself. Hence male driver alleles can be located at various loci of sex chromosomes since these generally do not recombine in case of the XY systems, but in case of autosomes and females, driver alleles are mostly restricted to the regions with limited recombination such as centromeres. Whereas male meiotic drive usually acts post-meiotically through sperm elimination and female meiotic drive acts during meiosis through exploiting the asymmetry of female division, the post-meiotic female drive and meiotic male drive have been reported previously. Another widely recognised form of meiotic drive in a broad sense is the meiotic drive in haploid spores in fungi, which works in a similar way as the male meiotic drive [91].

Current literature suggests that female meiotic drive is present in many (if not the majority) vertebrates from fish to mammals [91]. Indeed, more many mammalian karyotypes consist completely of chromosomes exhibiting exclusively monoarmed or biarmed morphology, whereas the combination of both is not as common [92]. It has been proposed that female meiotic drive is responsible for this, since it is capable of distorting segregation completely in favour of either morphology. Hence female meiotic drive is believed to drive karyotype evolution of these species, bringing the respective morphology to fully saturate a given karyotype (even though it has not been documented that this provides any selective advantage). Furthermore, there is always a possibility of switching from one state to another due to the spindle polarity reversal [93]. Similar non-random distribution of chromosome morphologies has been reported in certain orders of fishes, with female meiotic drive also being believed to fuel karyotype evolution of these clades [94]. Potential role of female meiotic drive has been proposed in birds for chicken chromosome 1 [88], as well as germline restricted chromosomes (GRCs) of songbirds [95], where these elements transmitted exclusively through females may selfishly exploit oocyte division asymmetry to remain in egg throughout meiosis. Indeed, the prevalence of a particular chromosome morphology (monoarmed or biarmed) is, just like in mammals, also common in invertebrates, such as entelegyne spiders [96]. Therefore, female meiotic drive may be more widespread phenomenon than previously anticipated across animals.

There are some well-documented instances of meiotic drive in plants, too. Perhaps the most iconic one is found in maize, where four knobs on so called abnormal chromosome 10 (Ab10) act as neocentromeres enhancing transmission of Ab10 (and also the rest of chromosomes containing knobs in the presence of Ab10) preferentially to megagametophyte (Table 1) [97]. Apart from neocentromeres, centromeres were thought to possibly act as driving elements already in 2001 [98,99,100], renaming the mechanism as centromere drive. However, this was not demonstrated in natural populations to explain meiotic drive until 2008 [101,102]. To understand the role of the centromere in the meiotic drive, F_1_ hybrid monkey flowers were generated by crossing *Mimulus nasutus* and *Mimulus guttatus*. The F_1_ hybrids contain D locus on homologous chromosomes, which is thought to be the centromeric region. Chromosome transmission rate was examined in *Mimulus nasutus* (maternal; dd) × *M. guttatus* (paternal; DD) F_2_ hybrids. According to Mendel’s first law, the expected ratio of D locus in F2 hybrids is Mendelian (1:2:1, dd:Dd:DD). Strikingly, there was a strong transmission bias against the *M. nasutus* allele—the actual ratio was extremely non-Mendelian (0:2:2, dd:Dd:DD) [102]. To prove the key role of D locus, F_2_ hybrids were repeatedly backcrossed with homozygous *M. nasutus*. Several generations later, it was shown that despite most of the genome originated from *M. nasutus*, the D locus from *M. guttatus* was still observed in over 90% of the offspring (Figure 4). Therefore, the extremely biased transmission indicates that the D locus could play a key role in transmission ratio distortion (Table 1 and Figure 4) [101,102].

Meiotic drive can cause biased segregation of homologous chromosomes when they segregate at anaphase I. Some organisms segregate homologs during meiosis II in a process called “inverted meiosis”, which is observed in groups possessing holokinetic chromosomes (i.e., chromosomes binding kinetochore microtubules usually along their entire length, e.g., worms). In these cases, meiotic drive operates during the second meiotic division. Mealybugs present an exemplary case—their chromosomes are holokinetic and indeed homologs segregate during anaphase II. Additionally, all paternal homologs are facultatively heterochromatic and only euchromatic (that is maternal) homologs bind to the spindle during male meiosis II—this is possible due to the formation of monopolar spindle, which binds euchromatic chromosomes only (Table 1). As a consequence, four haploid cells are produced, two containing paternal and two containing maternal homologs only. Those cells possessing paternal homologs only will not become functional sperm and they degenerate [104]. This example demonstrates that the meiotic drive can distort chromosome segregation dramatically, which can in turn lead to post-meiotic determination of which cells become functional sperm.

Molecular mechanisms underlying meiotic drive are still poorly understood and indeed different mechanisms might underpin the different examples of meiotic drive, but an increasing number of recent studies significantly contributes to this understanding, mainly in mouse oocytes. Suspected mechanisms and structures underpinning meiotic drive include following (summarized in Table 1): centromeres that can be classified as strong (low levels of Spc24 and Major satellite repeat content; high levels of HEC1 and Minor satellite repeat content; and CENP-A histone variant more expanded) or weak (levels of respective components are reverse). Since CENP-A is more expanded across strong centromeres compared to the weak ones, it may contribute to formation of larger kinetochores [11,105,106,107,108]. Furthermore, it seems that spindle polarity is established early after the spindle is formed and is still located in the middle of an oocyte. The cortical side of an oocyte is characterised by higher density of MTOCs and microtubules compared to the egg side and chromosomes possessing strong centromeres preferentially attach to the microtubules emanating from the egg side. If this is not the initial configuration (that is chromosomes with strong centromeres are initially attached to the cortical side microtubules), re-orientation occurs, regulated by Aurora kinases. Consequently, strong centromeres face the egg side and weak centromeres face the cortical side [106]. After the spindle migration occurs, concentration of Ran-GTP near cortex increases, possibly disturbing the initial Ran GTPase gradient established after nuclear envelope breakdown to promote spatial coordination of self-organisation of microtubule-chromosome system [52].

In terms of RanGTP formation, Ran guanosine exchange factor Regulator (Chromosome Condensation 1, RCC1) promotes the transformation of RanGDP to RanGTP, with chromosomes serving as sites of the RanGTP generation [53]. Diffusion from the chromosomes creates a decreasing RanGTP gradient from the nucleus to the cell cortex (Figure 5) [109]. Microtubule nucleation and stabilization are supported by RanGTP which activates critical spindle assembly factors (SAFs), such as TPX2 and NuMA, by releasing them from inhibitory Importin-α/β [52,110,111,112]. In mitosis, 22 spindle assembly factors were regulated by RanGTP, however, the role of RanGTP in SAFs regulation is still unclear in meiosis. In the cytoplasm, away from chromosomes, RanGTP is converted back to RanGDP by RanGTPase at the edge of RanGTP gradient. In such dynamic system, the size of spindle is considerably restricted [113].

The proximity of Ran-GTP to cortex in turn activates CDC42 signalling pathway leading to differential levels of tyrosination on egg and cortical microtubules. Ref. [105] argues that re-orientation of chromosomes (so that strong centromeres face the egg side and weak the cortical side) occurs at this point and is dependent precisely on microtubule tyrosination levels, in contrast to the result of [106] who claims that this re-orientation occurs before spindle migration and is dependent on Aurora kinase and the asymmetry of MTOCs distribution (Figure 6). The latter study was conducted on normal mouse cytotypes (possessing acrocentric chromosomes but differing in levels of mentioned centromeric and kinetochore components between two various strains) and thus it is applicable more generally, whereas the former study focused on heterozygotes for Robertsonian translocation. Nevertheless, the results of both studies indicate that meiotic drive is a multifactorial multistep process on molecular level. It would be interesting to look further at the connection between Aurora kinase-dependent chromosome re-orientation and spindle assembly checkpoint (SAC), since Aurora kinases are involved in both processes, possibly indicating that SAC acts as a mechanism preventing meiotic drive.

Molecular details in other driver systems are known less, but they also offer pieces of this very complex puzzle. The recent study in maize highlights involvement of molecular motors—in plants possessing Ab10 chromosome, meiotic drive is impaired without the functional KINDR region, which is a cluster containing genes coding for kinesin motor that binds exclusively to specific 180 bp repeat-containing knobs (Table 1) [97]. The presence or absence of meiotic drive has been also investigated in another plant, *Luzula*—it was found that meiotic drive does not operate in this case, which authors believe may be caused by holokinetic structure of chromosomes in this plant *per se* [114]. However, as was described above, meiotic drive in mealybugs operates indeed and it does through the monopolar spindle. On the other hand, unlike *Luzula*, mealybugs exhibit inverted meiosis and peculiar heterochromatinization of the entire homolog set [104]. Taken together, these data indicate that mechanisms through which meiotic drive can cause biased segregation in organisms possessing monocentric, but also holokinetic chromosomes, might involve different components. However, these may possibly be just parts of more complex yet similar pathways under control of same or similar genes that still need thorough future investigation—indeed, it has been demonstrated that in mitotically dividing *Drosophila* germline stem cells, mechanisms similar to the meiotic drive occur in order to segregate sister chromatids non-randomly in relation to the cell fate (where one cell stays the stem cell in order to preserve its infinite replication potential and the other cell will terminally differentiate). This includes temporal spindle asymmetry or quantitative centromere asymmetry, leading authors to name this phenomenon mitotic drive [115].

Biased segregation of autosomes can be seen if there are differences between homologous autosomes, but not necessarily between produced daughter cells. Indeed, such segregation bias has been found in *Caenorhabditis elegans*—if homologs of an autosome pair differ in size and sequence (due to the experimental insertion or deletion of a large transgene, or even smaller indels), the smaller homolog segregates with X chromosome and the larger against it [103]. This skew was later also found to be the ancestral trait of the entire *Caenorhabditis* lineage [116]. However, it should be noted that chromosomes of this worm are holocentric and similar situation has not been reported in organisms with monocentric chromosomes so far. Furthermore, the question remains whether this situation could be observed in the presence of more than just a single sex chromosome (for example XX/XY system), or it is just an artefact of the XX/X0 system of this animal. Nevertheless, this example shows the true non-random segregation of chromosomes, where certain chromosomes segregate together relative to each other and, unlike meiotic drive, this is not related to the cell fate.

### 3.2. Non-Random Segregation of Sex Chromosomes

In heterogametic XY males, X and Y chromosomes always segregate randomly because there are no differences between produced sperm cells, unless male meiotic drive (or some other mechanism) operates that can cause functional differences between sperm cells (e.g., only sperm containing X driving chromosome are functional, whereas Y-bearing sperm are impaired or eliminated). However, this is still not non-random segregation of sex chromosomes, but rather non-random transmission of gametes, since the functionality of respective sperm cells is established post-meiotically, based on which sperm contains which sex chromosome. Likewise, no preferential segregation of a single sex chromosome pair together with specific autosomes has ever been reported. The situation changes, however, as soon as there are more than one X or Y chromosomes (or both) present—this is because in systems with multiple sex chromosomes, all X chromosomes must normally segregate together to one pole, whereas all Y chromosomes together to another, which is obviously a non-random process itself and requires a regulation system. 

An example of early observations is the mole cricket *Neocurtilla hexadactyla*, an organism possessing a X_1_X_2_Y sex chromosome system. The original study incorrectly identified the X_2_Y bivalent as a heteromorphic autosome pair, while claiming that the remaining univalent represents the X_0_ sex chromosome system. Nevertheless, the key observation was made that the larger autosome of the heteromorphic pair always segregates with the remaining X chromosome and the smaller autosome always against it [117]. This biased segregation was later confirmed to be the segregation of X_1_X_2_Y system—indeed, the large autosome is the X_2_ and the smaller one the Y [118]. This example shows that the segregation of a multiple sex chromosome system appears to be naturally non-random, as the segregation of X_2_Y bivalent always occurs so that X_2_ (and not Y) segregates together with X_1_. How exactly such form of non-random, regulated segregation is achieved, however, remains a mystery. Likewise, many mammalian species possess X_1_X_2_Y neo-sex chromosome system and even more XY_1_Y_2_ [119] and again, it is unknown how the regulated segregation occurs in these systems. Yet another even more mind-puzzling example is found in heterogametic males of organisms that stably keep their multiple sex chromosomes in form of ring or chain multivalents, or so-called chromosome multiples. Surprisingly, not much research has been done regarding behaviour and segregation pattern of sex chromosome multiples for decades, even though the presence of such multiples has been reported in a number of species [120]. Notably, some recent research has been done in frogs [121,122]. Nevertheless, the behaviour and segregation of sex chromosome multiples, as well as respective molecular mechanisms, are still poorly understood. Basically, a sex chromosome multiple can form a chain, ring, or even more complex multivalent structure to achieve regulated segregation. It is widely accepted that in case of chain multiples, the alternate segregation of Xs and Ys is the key. For that, a chain must take a zig-zag orientation (Figure 7), as is the case in Australian huntsman spider *Delena cancerides*—the species harbouring populations with sex chromosome chains involving 3–19 chromosomes or even populations with two distinctive chains [123,124,125]. Similarly, a peculiar chain of nine or ten sex chromosomes can be found in echidna and platypus, respectively—a situation unique to mammals [126,127]. In these organisms (and many others possessing long sex chromosome chains, such as termites [128]), it is unknown what exactly is responsible for the formation of necessary zig-zag configuration resulting to alternate segregation. Several factors have been proposed to be involved, including chiasma or centromere position, or chromosome size. However, these factors do not apply to at least mentioned *D. cancerides* [120], so even if they have some contribution, they are not universal and definite. In addition to that, it has been shown that the assembly of platypus sex chromosome chain during male prophase I is a highly organised pairing process [129] and involves differential cohesin loading to paired and unpaired region of sex chromosomes. Specifically, SMC3 subunit of cohesin complex is massively recruited to unpaired regions of sex chromosome chain during zygotene [130]. It has been shown that SMC3 is required for synapsis of homologs and the assembly of synaptonemal complex might begin precisely at euchromatic sites where SMC3 is most abundant—indeed, knockdown of both SMC1 and SMC3 leads to a complete block of synapsis in *Drosophila* oocytes, demonstrating that SMC3 is a critical component in the formation of synapsis [66,131]. Therefore, the presence of cohesins is critical to complement the function of synapsis [132]. On the other hand, sex chromosomes in platypus are non-homologous except for PARs and therefore do not undergo synapsis, yet still accumulate SMC3. Indeed, [130] speculate that this finding might be ascribed to SMC3 roles specific for alternate segregation of this complex sex chromosome chain. For many of these examples we still entirely lack an understanding of the mechanisms underpinning this type of non-random segregation. Selection for ability to segregate alternately has been demonstrated in plants [120]. Our current best leads might therefore be the genetic control and perhaps molecular composition and organisation of components responsible for chromosome segregation. This could include centromeres and kinetochores (with centromere orientation potentially being responsible for the zig-zag orientation of multiples) or cohesins having the very specific function especially in case of largely achiasmatic sex chromosomes.

In case a sex chromosome multiple forms a ring, it also needs to segregate alternately. It is easier to imagine alternate segregation of a more simple ring composed of four sex chromosomes—the alternate (and not adjacent) segregation could in theory mechanistically occur if a ring takes a shape of figure eight (∞), a configuration equivalent to the zig-zag shape (Figure 8). However, it is unknown what would be responsible for formation of such orientation and what is more, it is harder to imagine this to be the case in rings of more than four sex chromosomes (that is the formation of a repeated figure eight with chromosomes folding over each other multiple times), such as in Amazonian frog *Leptodactylus pentadactylus* containing stable ring multiple composed of six X and six Y chromosomes [121]. Finally, sex chromosome multiples can form with even more complex multivalents, which has been reported for example in charcoal spider *Tegenaria ferruginea* [133] or butterfly *Leptidea amurensis* [134]. In the former, attachment plaques and centrosome are both involved in the pairing of X_1_X_2_X_3_X_4_X_5_Y hexavalent, which points out to the possible involvement of these structures in regulated segregation. In the latter, the presence of ZW multiple itself is interesting (there are no such multiples reported in birds also exhibiting the ZW system, which is the system where females carry heterogametic ZW chromosomes as opposed to heterogametic males of the XY system), but perhaps not so surprising given the holokinetism of butterfly chromosomes. It has been proposed that multiple sex chromosome systems in heterogametic females could be subject to female meiotic drive and in turn lead to biased sex ratio, contrary to Y chromosome in heterogametic males, which is obviously always shielded from effects of female meiotic drive [135]. It has also been proposed that the monotreme multiple XY system could have evolved from a ancestral amniote ZW system [120,136], however more recent comparative sex chromosome analysis supports the independent origin of monotreme, avian and eutherian sex chromosomes [137]. Is it therefore possible that the switch from female to male heterogamy is required for existence of sex chromosome multiples in organisms with monocentric chromosomes? If so, it even further supports the necessity to investigate possible molecular differences between segregation components of X and Y chromosomes in systems possessing sex chromosome chains such as platypus in order to shed more light on regulated segregation of sex chromosomes. Furthermore, it is now known that multivalents involving sex chromosomes are not restricted to heterogametic males only—thanks to the presence of special feminising X chromosome (X*) in African pygmy mouse involved in whole arm reciprocal translocations, there are three female genotypes, two of which (XX* and X*Y) involve a quadrivalent [138]. It would be of great interest to look more into such unusual situation in regard to chromosome segregation, too.

## 4. Concluding Remarks

The results of Eleanor Carothers combined with basic Mendelian principles led to the accepted model that segregation of autosomes is by default random and that observations like meiotic drive are an exception and evolved either in selfish elements or as means to eliminate genetic information (such as B chromosomes found for example in plant root cells). There is a growing number of examples of chromosomes violating random segregation. Here we outline these examples and the molecular and cellular mechanisms that may be involved. More recent work shows that genetic alteration of chromosomes can affect segregation with sex chromosomes adding a new aspect to cytological forces affecting segregation. Together these effects enable transition of random to non-random segregation and the extent of non-random segregation depends on a number of genetic, epigenetic, and cellular mechanisms. If this is the case, it may have profound effects on the evolution of chromosomes and sex chromosomes. Perhaps biased segregation of already heteromorphic autosomes can fuel their transition into sex chromosomes? Indeed, a particular autosome pair in the white-throated sparrow contains homologs that differ in the locus containing a supergene cluster consisting of more than thousands of genes (referred to as chromosomes 2 and 2*^m^*). This in turn causes that such autosomes behave as de facto sex chromosomes given the lack of recombination, almost exclusive disassortative mating between individuals containing different homologs and that the chromosome 2*^m^* is almost never present in homozygous condition. In other words, different supergene clusters of this heteromorphic chromosome pair cause the non-random mating between individuals of this bird species, just like sex chromosomes, because the chromosome 2 behaves as a Z chromosome and the chromosome 2*^m^* as a W chromosome. Moreover, it seems that 2*^m^*, much like W or Y chromosomes, degrade gradually due to the lack of recombination and accumulation of deleterious mutations in the evolution of these sex chromosomes. Indeed, authors ascribe this to the presence of different supergene clusters [139]. Perhaps similar molecular mechanisms (suppression of recombination and degradation) shared between sex chromosomes and supergene clusters together with non-random segregation of autosomes can be linked with the origin and evolution of sex chromosomes. This may be through supergene clusters promoting non-random segregation or acquiring sexually antagonistic genes linked with disassortative mating.

Sex chromosomes, unlike most autosomes, evolve rapidly into heteromorphic chromosomes [140]. If there are more X or Y chromosomes (or both), all Xs and all Ys must segregate together respectively and never in a mixed fashion, an extreme example of non-random segregation. This is achieved by alternate segregation of sex chromosomes, but it is still unknown what would be the cause of this. Perhaps the answer lies in various mechanisms and components we can see occurring during meiotic drive, for example all X chromosomes having one type of centromere and all Y chromosomes having the opposite type, which could lead to differences in kinetochores. This might in turn secure that all X chromosomes are connected to microtubules emanating from one pole of a cell, whereas all Y chromosomes to microtubules emanating from the opposite pole and this itself might be sufficient. Another possible scenario involves formation of a zig-zag structure of multiple sex chromosomes chains (or figure eight shape of circular multiples) occurring—in this case, cohesins might be involved as proposed by studies in platypus [130]. A better understanding of the genetic and epigenetic aspects of centromeres would be a first step to explore their role in alternating segregation.

Chromosome segregation can occur in different ways ranging from random to non-random and alternating segregation of both autosomes and sex chromosomes. The critical question remains whether these types of segregation share similar molecular mechanisms (and components), or if they are fundamentally different. Novel insights into the genetic, epigenetic and cytoplasmic factors underpinning different types of chromosome segregation will advance our understanding of karyotype evolution but may also play a major role in the evolution of sex chromosomes, which must always evolve from what was once an autosome.

## Figures and Tables

**Figure 1 genes-12-01338-f001:**
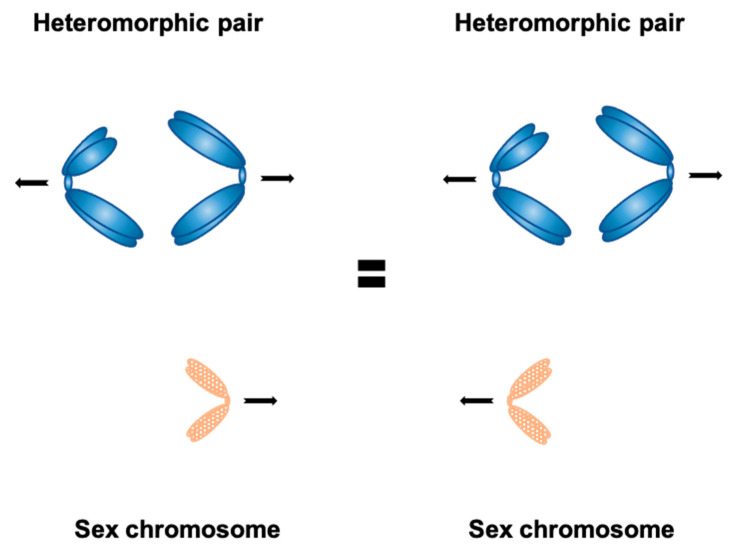
Carothers’ observation in the grasshopper *Brachystola*—the frequency of accessory (sex) chromosome segregation together with either chromosome belonging to the heteromorphic pair is roughly the same; this led to conclusion that chromosome segregation is generally random.

**Figure 2 genes-12-01338-f002:**
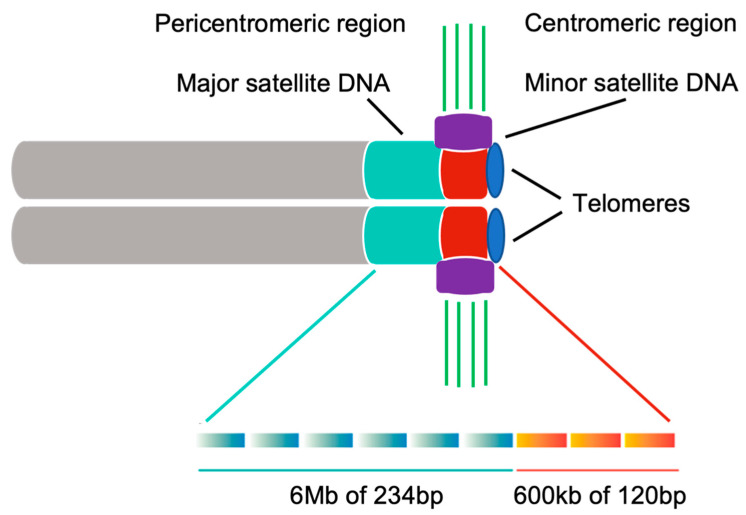
Major satellite repeats are localized at the pericentromeric region. Minor satellite repeats are localized at the centromeric region. Rectangular gradient shows the monomer for each satellite repeat. The kinetochore is mainly recruited to minor satellite DNA and attached by microtubules directly. Cyan: major satellite DNA; Red: minor satellite DNA; Purple: kinetochore; Green: microtubules; Gray: sister chromatids of a mouse chromosome; Blue: telomeres.

**Figure 3 genes-12-01338-f003:**
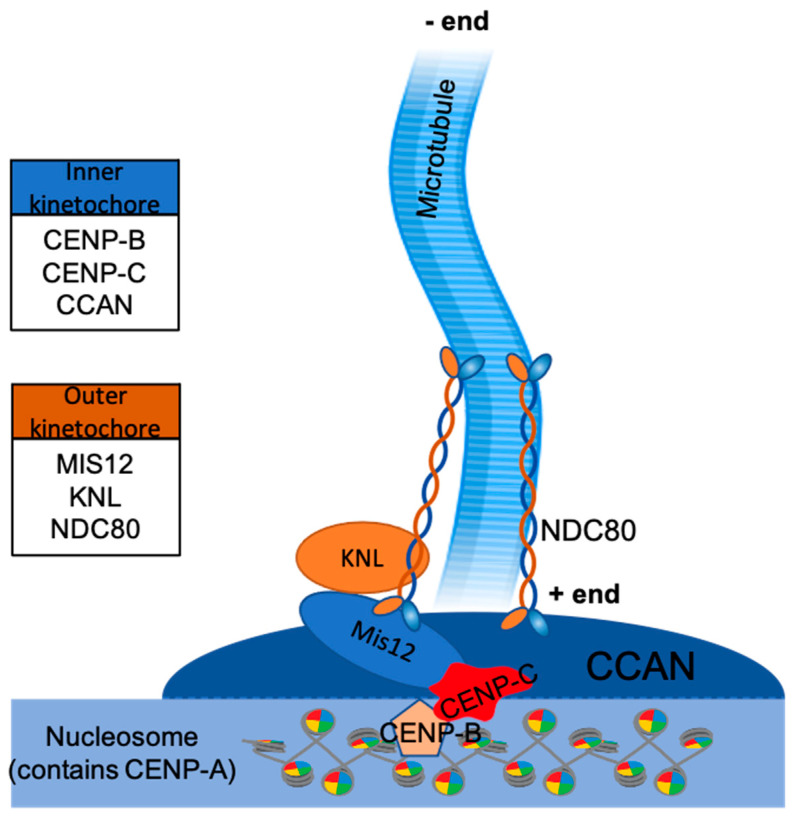
Proteins of KMN complex (KNL, Mis12, and NDC80) belong to the outer kinetochore layer, proteins of constitutive centromere-associated network (CCAN), CENP-B and CENP-C belong to the inner kinetochore layer. Together they form the required foundation for kinetochore-microtubule interface. Histone H3 is replaced by CENP-A belonging to the nucleosome. Microtubules attach to kinetochores at their plus ends, with minus ends being anchored to a microtubule organising centre (MTOC).

**Figure 4 genes-12-01338-f004:**
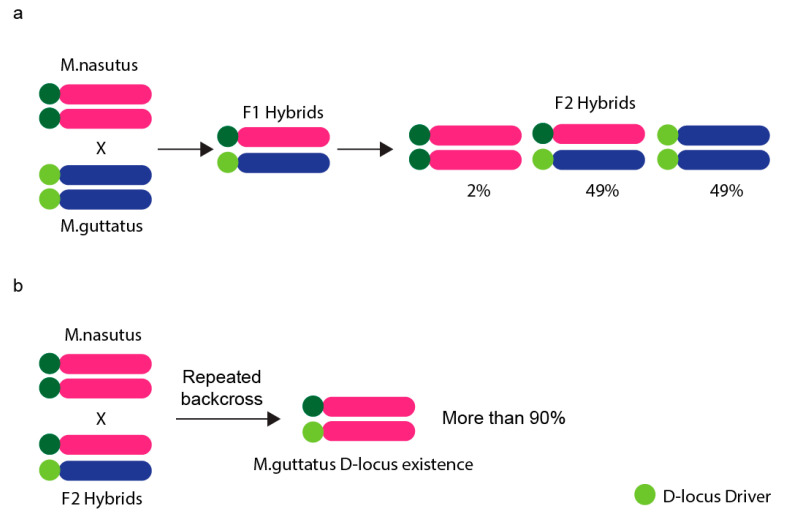
Biased chromosome transmission in hybrid Monkeyflowers. (**a**) F_2_ hybrid Monkeyflower crosses of *Mimulus nasutus* (maternal; dd) and *Mimulus guttatus* (paternal; DD) were used to examine the chromosome transmission rate. The chromosome from *M. guttatus* with D-locus is preferentially transmitted to the next generation. (**b**) Female F_2_ hybrids were repeatedly backcrossed with male *M. nasutus* and the genome was purified to the homozygous state for *M. nasutus* alleles after several recombination events. However, over 90% of D-locus from *M. guttatus* remained in the offspring.

**Figure 5 genes-12-01338-f005:**
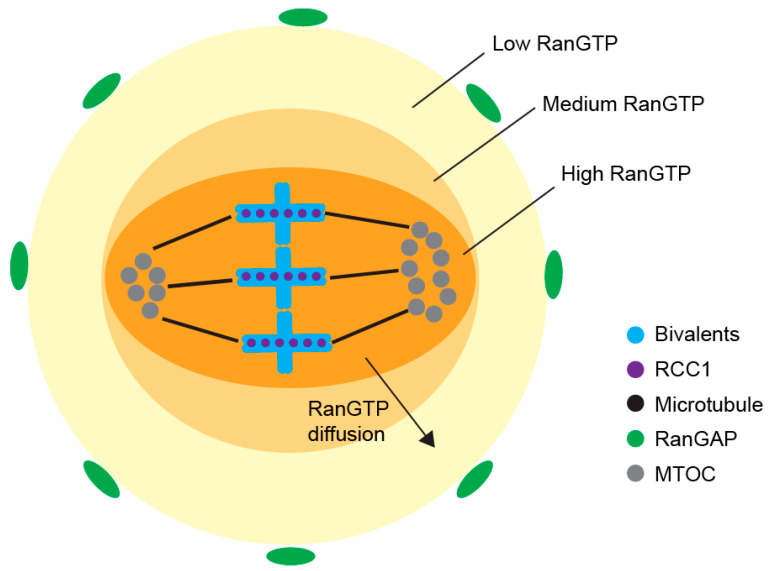
RanGTP gradient around meiosis I spindle—at the chromosomes, RCC1 converts RanGDP to RanGTP and diffusion of RanGTP forms the RanGTP gradient. In the cytoplasm, RanGAP converts RanGTP back to RanGDP, setting the boundary for the RanGTP gradient, and defining the region for spindle formation in the oocyte.

**Figure 6 genes-12-01338-f006:**
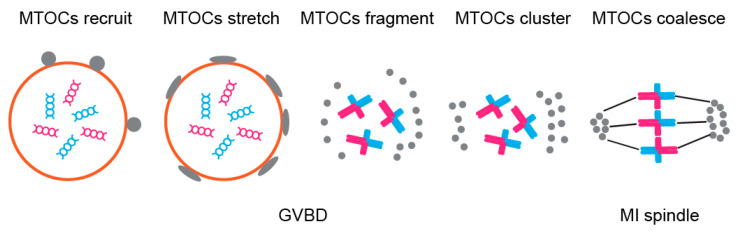
Asymmetrical MTOCs formation—MTOCs (grey) are recruited to nuclear envelope and are well stretched before NEBD. After NEBD, MTOCs are fragmented to small pieces and migrate to spindle poles. Then the MTOCs cluster to form spindle poles and nucleate microtubules in order to assemble the spindle.

**Figure 7 genes-12-01338-f007:**
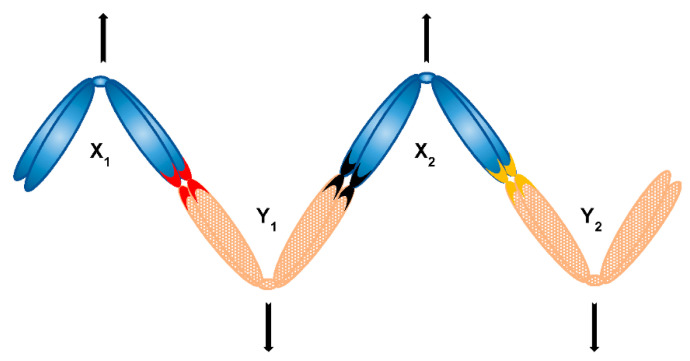
Alternate segregation of a chain sex chromosome multiple consisting of four sex chromosomes (X_1_X_2_Y_1_Y_2_ system) can be achieved with formation of a zig-zag structure; respective contacts of sex chromosomes (e.g., PARs) are depicted as coloured chromosome tips.

**Figure 8 genes-12-01338-f008:**
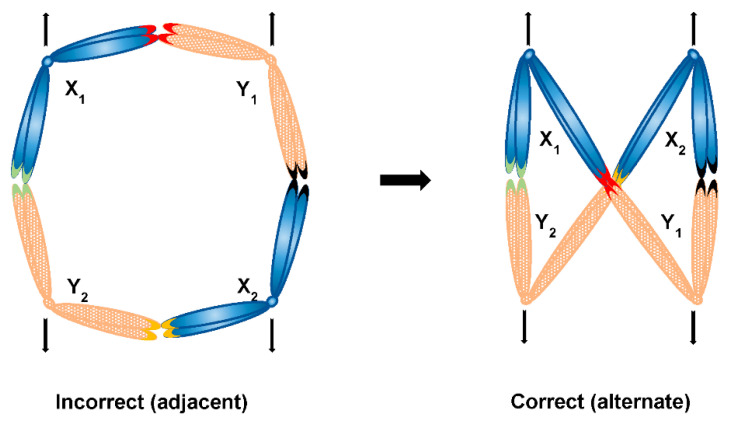
Possible segregation modes of a ring sex chromosome multiple consisting of four sex chromosomes (X_1_X_2_Y_1_Y_2_ system)—in case of a plain ring, segregation of sex chromosomes would be adjacent, which would in turn lead to the incorrect segregation of X and Y chromosomes together; in case of a figure eight shaped ring, alternate segregation is achieved leading to the correct segregation of Xs together and Ys together respectively; respective contacts of sex chromosomes (e.g., PARs) are depicted as coloured chromosome tips.

**Table 1 genes-12-01338-t001:** Selected known examples of the meiotic drive, including their mechanisms.

Example of Meiotic Drive	Male/Female Drive	Components and Mechanisms Involved	Reference
*Caenorhabditis elegans*	N/A	Size heteromorphism	[103]
*Drosophila*	Male	*Sd* driver inducing sperm elimination (histone-to-protamine transition failure)	[90]
Maize	Female	Ab10 knobs (not linked to centromeres), KINDR kinesin motor	[97]
Mealybug	Male	Monopolar spindle	[104]
Monkeyflower	Female	Centromere-based mechanism (involving diversified repetitive DNA?)	[102]
Mouse	Female	Weak and strong centromeres	[105,106]
Mouse	Male	*t* driver impairing embryonic development and sperm motility	[90]
Songbirds	Female	Polarized recombination pattern of germline restricted chromosomes?	[95]

## Data Availability

Not applicable.

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
