# Peer review of "Flavors of Non-Random Meiotic Segregation of Autosomes and Sex Chromosomes"

_genes, 2021, doi:10.3390/genes12091338_

Round 1

Reviewer 1 Report

The article “Flavors of non-random segregation of autosomes and sex chromosomes” by Filip Pajpach, Tianyu Wu, Linda Shearwin-Whyatt, Keith Jones and Frank Grützner focuses on the evidence for biased chromosome segregation. The publication of such a review is important, especially to draw attention to the existence of different mechanisms of non-random segregation of gonosomes and autosomes in meiosis and their role in linking traits. Although the hypotheses expressed have been published before in disparate works, the text of the text is original and meaningful for overcoming stereotypes. The manuscript corresponds to the scope of “Genes” and it should be of interest to the readers of Special Issue: Sex Chromosome Evolution and Meiosis.

The article is intended to be included in the thematic subspecialised issue, which explains the lack of mention of meiosis in the title of the article. However, this is a drawback, as it can be misleading in an independent search. This is all the more important to point out because the question of non-random chromosome segregation during mitotic divisions has been the subject of extensive debate (see Nagel et al. and the opposing position in Solovei et al.).

Line 86-133. A more detailed description of the structure of centromeric regions of chromosomes is required, indicating epigenetic differences between centromeric repeats and pericentromeric heterochromatin. it is also important to mention the role of CENP-A in centromeric interference preventing recombination in the regions of these repeats.

Line 268-293. The section on the role of cohesins in segregation is very abstract and could be expanded to include work on segregation disruption in the first and second divisions of the meiosis done in mice and to use data on such disruption in humans. A description of the role of synapsis in chromosome segregation in the meiosis is completely missing. Description of asynaptic mutant disorders in both plants and animals.

The role of pachytene arrest in homologous synapsis disruption and the existence of a special mechanism for protecting the unpaired regions of heteromorphic sex chromosomes are important to point out (sex XY vesicle in mammals, corkscrew heteromorphic ZW bivalent in birds). In this regard, the interchromosomal effect and biased segregation of autosomes and sex chromosomes when the position of the sex vesicle in the nucleus architecture is changed should be mentioned further in the paper.

Among the examples of non-random segregation, it is important to mention germline restricted chromosomes (GRC). Their key difference from B chromosomes is in their segregation features.

Line 610. Refers to the phrase “such as B chromosomes in plant root cells” – B chromosomes are not restricted to plant root cells.

Author Response

Dear Reviewer 1:

We are grateful for the positive and constructive comments  and we are pleased to submit a revised manuscript where we have taken on almost all the suggested changes (detailed below):

REVIEWER 1

Line 2. In the title, I suggest to add that they are talking about meiosis, since there is no evidence in the manuscript that mitosis is discussed (example: “non-random meiotic segregation”).

• We changed the title by adding “meiotic” to better reflect the content of the manuscript and special issue.

Line 48. It would it advisable to add here that the "dogma" was formulated over time also because it fits very well with the random segregation of single genes located on non- homologous chromosomes (i.e., Mendel’s laws), something that was more easily (and in earlier times) demonstrated than physical.

• We clarified that the dogma of random chromosome segregation fits Mendelian rules in the context of historical notes.

Line 53. Fix “Drosophila”. Line 179. Check spelling (“turned”).
• We fixed typos and improved punctuation and readability.

Line 62*. A fundamental step of accurate meiotic chromosome segregation is homologous recombination - this aspect is not illustrated and discussed in the manuscript. See for example doi: 10.1146/annurev.genet.36.041102.1139. Authors are asked to fill this gap, or to explain why this aspect is not reported and discussed in the review.

• We added a section involving homologous recombination and synapsis, which also expands the cohesion section. We included description of asynaptic mutants (both cohesins and synaptonemal complex proteins) and how it relates to chromosome segregation.

Line 81*. For the words “nucleic acid”, it is better to write here about nucleic acids (plural) since centromeres harbor both DNA and RNA (see for example doi: 10.1016/j.jmb.2020.03.027 and doi: 10.3389/fmolb.2021.642732 but there are many others); also RNA is important for proper spindle attachment. The authors are asked to report and discuss the role of RNA in centromere function.

• We reported and discussed the role of RNA in centromere function.
Line 107. it would be advisable to add a small section about neocentromeres and holocentromeres, and their evolutionary role in species formation and disease (cancer). See for example doi: 10.3390/genes11070809 ; doi: 10.1016/j.ajhg.2007.11.009 ; doi: 10.1007/s10577- 012-9296-x. These structures are also discussed later, but an introduction here on their general organization (a few lines) would help the non specialist reader.

• We added a section about neo-centromeres and holocentromeres and their relevance.

Line 162. Add the words “outer kinetochore” and “microtubule” to Figure 3. Line 166. This list of proteins (down to line 171) without the possibility to see them in the Figure 3 may be confusing for non experts. Is it possible to make a more comprehensive picture? Line 186. +/- microtubule ends should be indicated in figure 3, too.

• We improved Figure 3 based on the provided comments (added the requested key words and made it more comprehensive).

Line 181. Instead of “far from stable” I'd say "dynamic" since this allows a fine tuning of chromosome movement. Line 305. “this is often”: despite the non negligible frequency, I believe that "often" is too strong - this is still largely considered not the general case, as also reported by the authors in their concluding remarks. Please revise.

• We changed “far from stable” to “dynamic” as suggested and addressed similar changes as well (i.e., word “often” in the context of non-random segregation on the original line 305).

Line 185. Regarding the putative ring structure, this is not explained before; better to explain the 3D organization of the kinetochore and, if possible, make it evident in figure 3.

• We clarified the putative ring structure in the context of conformational wave microtubule model.

Line 207. Regarding Dumont et al. (2007), please formalize the citation using square brackets. • We corrected the citation to match the format with the rest.

Line 210. “laterally”: for non experts, this should be better explained and/or illustrated in a figure.

• We clarified the lateral attachment of microtubules to kinetochores.
Line 313*. Because of the extensive description of female meiotic drive after this point, here it would be appropriate to describe the male drive as well, and to make a comparison between the two mechanisms (similarities and differences). Moreover, it would be appropriate to highlight, in the following examples, if the phenomena occur in males or females (a modification of Table 1 would suffice). Finally, all examples reported should be included in Table 1 (Drosophila and C. elegans are missing, for example, as well as many others). A comprehensive Table 1 would help non expert readers to track the reported information.

• We described male meiotic drive better, highlighting similarities and differences with female meiotic drive and I expanded Table 1, so it now includes all examples of meiotic drive discussed in the review as well as indicating whether it is male or female drive.

Line 357. As for the second M. nasutus, shouldn't this be M. guttatus?
• We corrected the second M. nasutus to M. guttatus on the original line 357.

Line 374. “However”: this suggest a different output of meiotic drive in inverted meiosis, which is not. Please reword.

• We reworded this section, so it does not look like the outcome of meiotic drive is different compared to the classic meiosis.

Line 577. For non experts, the ZW system should be briefly described - one sentence at least, just to fix the difference with its XY counterpart, in terms of heterogametic sex, to introduce the following descriptions.

• We introduced the ZW sex chromosome system better.
Line 629. “degrade gradually”: please define “degradation” for those who are not into sex chromosome decay.

• We defined the degradation of sex chromosomes in terms of the classic model of the sex chromosome evolution.

Line 630. Regarding the evolution of supergene clusters, is this the author's thought or it is derived from literature data? Please explain and add references, if appropriate.

• We clarified this part to make it clear what is derived from literature data and what are our own thoughts.

Line 633. “evolve rapidly into heteromorphic chromosomes”; please add reference(s). • We added the appropriate reference.

Line 498. An history note about the experiments by Bridges and Morgan (1916) on Drosophila would be appropriate here.

• We considered adding this section where suggested or somewhere else in the review, but felt that because these experiments related to location of genes on chromosomes, as confirmed by non-disjunction studies of Bridges and Morgan, do not fit very well within the scope and context of the review as they do not directly involve segregation of chromosomes but happy to consider this again if the reviewer feels strongly about including this aspect.

Reviewer 2 Report

In the manuscript genes-1320611 the authors report on the mechanisms of chromosome segregation (mainly, in meiosis) and provide an updated overview of the topic. The subject of the manuscript is intrinsically interesting, and in recent years only a minority of manuscripts report on chromosome segregation, especially from a cytological point of view. Thus, I believe that the manuscript is also timely. Overall, the quality of the work is very good, although there is some missing information (in some cases, this is a major point) to be fixed before publication. Below, my observations according to line number order. Major issues are indicated by an asterisk next to line number. For simplicity, I also report here the line numbers with major issues: 62; 81; 313.

Line 2. In the title, I suggest to add that they are talking about meiosis, since there is no evidence in the manuscript that mitosis is discussed (example: “non-random meiotic segregation”).

Line 48. It would it advisable to add here that the "dogma" was formulated over time also because it fits very well with the random segregation of single genes located on non-homologous chromosomes (i.e., Mendel’s laws), something that was more easily (and in earlier times) demonstrated than physical chromosome movement.

Line 53. Fix “Drosophila”.

Line 62*. A fundamental step of accurate meiotic chromosome segregation is homologous recombination - this aspect is not illustrated and discussed in the manuscript. See for example doi: 10.1146/annurev.genet.36.041102.1139. Authors are asked to fill this gap, or to explain why this aspect is not reported and discussed in the review.

Line 81*. For the words “nucleic acid”, it is better to write here about nucleic acids (plural) since centromeres harbor both DNA and RNA (see for example doi: 10.1016/j.jmb.2020.03.027 and doi: 10.3389/fmolb.2021.642732 but there are many others); also RNA is important for proper spindle attachment. The authors are asked to report and discuss the role of RNA in centromere function.

Line 107. it would be advisable to add a small section about neocentromeres and holocentromeres, and their evolutionary role in species formation and disease (cancer). See for example doi: 10.3390/genes11070809 ; doi: 10.1016/j.ajhg.2007.11.009 ; doi: 10.1007/s10577-012-9296-x. These structures are also discussed later, but an introduction here on their general organization (a few lines) would help the non specialist reader.

Line 162. Add the words “outer kinetochore” and “microtubule” to Figure 3.

Line 166. This list of proteins (down to line 171) without the possibility to see them in the Figure 3 may be confusing for non experts. Is it possible to make a more comprehensive picture?

Line 179. Check spelling (“turned”).

Line 181. Instead of “far from stable” I'd say "dynamic" since this allows a fine tuning of chromosome movement.

Line 185. Regarding the putative ring structure, this is not explained before; better to explain the 3D organization of the kinetochore and, if possible, make it evident in figure 3.

Line 186. +/- microtubule ends should be indicated in figure 3, too.

Line 207. Regarding Dumont et al. (2007), please formalize the citation using square brackets.

Line 210. “laterally”: for non experts, this should be better explained and/or illustrated in a figure.

Line 305. “this is often”: despite the non negligible frequency, I believe that "often" is too strong - this is still largely considered not the general case, as also reported by the authors in their concluding remarks. Please revise.

Line 313*. Because of the extensive description of female meiotic drive after this point, here it would be appropriate to describe the male drive as well, and to make a comparison between the two mechanisms (similarities and differences). Moreover, it would be appropriate to highlight, in the following examples, if the phenomena occur in males or females (a modification of Table 1 would suffice). Finally, all examples reported should be included in Table 1 (Drosophila and C. elegans are missing, for example, as well as many others). A comprehensive Table 1 would help non expert readers to track the reported information.

Line 357. As for the second M. nasutus, shouldn't this be M. guttatus?

Line 374. “However”: this suggest a different output of meiotic drive in inverted meiosis, which is not. Please reword.

Line 498. An history note about the experiments by Bridges and Morgan (1916) on Drosophila would be appropriate here.

Line 577. For non experts, the ZW system should be briefly described - one sentence at least, just to fix the difference with its XY counterpart, in terms of heterogametic sex, to introduce the following descriptions.

Line 629. “degrade gradually”: please define “degradation” for those who are not into sex chromosome decay.

Line 630. Regarding the evolution of supergene clusters, is this the author's thought or it is derived from literature data? Please explain and add references, if appropriate.

Line 633. “evolve rapidly into heteromorphic chromosomes”; please add reference(s).

Author Response

Dear Reviewer 2

We are grateful for the positive and constructive comments and we are pleased to submit a revised manuscript where we have taken on almost all the suggested changes (detailed below):

Line 268-293. The section on the role of cohesins in segregation is very abstract and could be expanded to include work on segregation disruption in the first and second divisions of the meiosis done in mice and to use data on such disruption in humans. A description of the role of synapsis in chromosome segregation in the meiosis is completely missing. Description of asynaptic mutant disorders in both plants and animals.

• We added a section involving homologous recombination and synapsis, which also expands the cohesion section. We included description of asynaptic mutants (both cohesins and synaptonemal complex proteins) and how it relates to chromosome segregation.

Line 86-133. A more detailed description of the structure of centromeric regions of chromosomes is required, indicating epigenetic differences between centromeric repeats and pericentromeric heterochromatin. it is also important to mention the role of CENP-A in centromeric interference preventing recombination in the regions of these repeats.

• We added a more detailed description of centromeres, highlighting epigenetic differences between centromeres and pericentromeres. We also mentioned the role of CENP-A protein in centromeric interference of recombination.

Among the examples of non-random segregation, it is important to mention germline restricted chromosomes (GRC). Their key difference from B chromosomes is in their segregation features.

• We added a mention of germline restricted chromosomes (GRCs).
Line 610. Refers to the phrase “such as B chromosomes in plant root cells” – B chromosomes are not restricted to plant root cells.

• We corrected this statement so it does not look like we are claiming that B chromosomes are only restricted to plant root cells.

The role of pachytene arrest in homologous synapsis disruption and the existence of a special mechanism for protecting the unpaired regions of heteromorphic sex chromosomes are important to point out (sex XY vesicle in mammals, corkscrew heteromorphic ZW bivalent in birds). In this regard, the interchromosomal effect and biased segregation of autosomes and sex chromosomes when the position of the sex vesicle in the nucleus architecture is changed should be mentioned further in the paper.

• This is an interesting topic and we expanded the section on synapsis and asynapsis (page 9 first section). This also covers some of the differences in sex chromosome pairing configurations. We are aware of the pseudopairing of chicken ZW (we suspect the corkscrew configuration is equivalent to what has also been referred to a zeta configuration) and literature around MSCI. However we feel that this is a topic that has been covered more recently and would make this section substantially larger and we could not find much original work to suggest how this relates to the topic. Similarily the comment on interchromosomal effects is very interesting but we could not find much research on this topic, except for clinical consequences of the interchromosomal effect that did not seem to fit well within the scope of the review. We'd certainly happy to consider this further and would be grateful for the reviewer to point us ti specific papers that are relevant.

Round 2

Reviewer 1 Report

The title of the article by Filip Pajpach, Tianyu Wu, Linda Shearwin-Whyatt, Keith Jones and Frank Grützner has been clarified and now better reflects the content. It focuses on the evidence for biased chromosome segregation. The publication of such a review is important, especially to draw attention to the existence of different mechanisms of non-random segregation of gonosomes and autosomes in meiosis and their role in linking traits. The manuscript corresponds to the scope of “Genes” and it should be of interest to the readers of Special Issue: Sex Chromosome Evolution and Meiosis.

The authors expanded and improved the description of the organization of centromeric and pericentomeric regions and added information on their epigenetic markings. However, a merged (confused) description of these areas in terms of their organization has emerged (line 153-178). It should be clarified that the inactive chromatin corresponds to the pericentromeric region, whereas the kinetochore binding site is distinct and characterized by euchromatin markers. Transcription associated with carcinogenesis refers specifically to heterochromatin pericentromeric sequences, whereas transcription of centromeric elements proper is important for maintaining the functional structure of the kinetochore (for review see https://compcytogen.pensoft.net/article/51895/). A small adjustment is required in this section.

The authors supplemented the information on synapsis and this improved the completeness of the meiotic segregation charateristics. This is important because consideration of the synapsis of heteromorphic gonosomes raises the question of synapsis completion before the transition to diplotene and the risks of pachytene arrest.

I can agree with the authors that this problem may have not only an evolutionary but also a clinical significance and may require a separate review, although a mention of the mechanisms of sex vesicle (XY body) formation is clearly lacking in such a review (for review see https://www.frontiersin.org/articles/10.3389/fgene.2012.00112/full).

Author Response

Dear Reviewer 1,

We are grateful for additional positive and constructive comments and we are pleased to submit a revised manuscript where we have taken on the suggested changes (detailed below):

The authors expanded and improved the description of the organization of centromeric and pericentomeric regions and added information on their epigenetic markings. However, a merged (confused) description of these areas in terms of their organization has emerged (line 153-178). It should be clarified that the inactive chromatin corresponds to the pericentromeric region, whereas the kinetochore binding site is distinct and characterized by euchromatin markers. Transcription associated with carcinogenesis refers specifically to heterochromatin pericentromeric sequences, whereas transcription of centromeric elements proper is important for maintaining the functional structure of the kinetochore (for review see https://compcytogen.pensoft.net/article/51895/). A small adjustment is required in this section.

  • We clarified that centromere contains euchromatin markers and its transcription is responsible for the centromere functionality whereas pericentromere contains inactive chromatin and its transcription is suppressed to prevent transposon reactivation, carcinogenesis and disease. We referenced the suggested review.

The authors supplemented the information on synapsis and this improved the completeness of the meiotic segregation charateristics. This is important because consideration of the synapsis of heteromorphic gonosomes raises the question of synapsis completion before the transition to diplotene and the risks of pachytene arrest.

  • We agree that this was a very helpful suggestion and improved the quality of the review.

I can agree with the authors that this problem may have not only an evolutionary but also a clinical significance and may require a separate review, although a mention of the mechanisms of sex vesicle (XY body) formation is clearly lacking in such a review (for review see https://www.frontiersin.org/articles/10.3389/fgene.2012.00112/full).

  • We added a small section that includes the mechanism of sex body formation and included the suggested reference.

Reviewer 2 Report

The Authors properly responded to criticism/suggestions.

Author Response

 Dear Reviewer 2: Pleased to hear that the comments have been addressed and thanks again for excellent comments.